# Modulation of the In Vivo Inflammatory Response by Pro- Versus Anti-Inflammatory Intervertebral Disc Treatments

**DOI:** 10.3390/ijms21051730

**Published:** 2020-03-03

**Authors:** Carla Cunha, Graciosa Q. Teixeira, Cláudia Ribeiro-Machado, Catarina L. Pereira, Joana R. Ferreira, Maria Molinos, Susana G. Santos, Mário A. Barbosa, Raquel M. Goncalves

**Affiliations:** 1i3S—Instituto de Investigação e Inovação em Saúde, Universidade do Porto, 4200-135 Porto, Portugal; carla.cunha@ineb.up.pt (C.C.); graciosa.teixeira@uni-ulm.de (G.Q.T.); claudia.machado@i3s.up.pt (C.R.-M.); catarina.pereira@ineb.up.pt (C.L.P.); joana.ferreira@i3s.up.pt (J.R.F.); maria.c.m.molinos@gmail.com (M.M.); susana.santos@ineb.up.pt (S.G.S.); mbarbosa@i3s.up.pt (M.A.B.); 2INEB—Instituto de Engenharia Biomédica, Universidade do Porto, 4200-135 Porto, Portugal; 3ICBAS—Instituto de Ciências Biomédicas Abel Salazar, Universidade do Porto, 4050-313 Porto, Portugal

**Keywords:** disc degeneration, inflammation, biomaterials, nanocomplexes, anti-inflammatory

## Abstract

Inflammation is central in intervertebral disc (IVD) degeneration/regeneration mechanisms, and its balance is crucial to maintain tissue homeostasis. This work investigates the modulation of local and systemic inflammatory response associated with IVD degeneration/herniation by administration of PRO- versus ANTI-inflammatory treatments. Chitosan/poly-γ-glutamic acid nanocomplexes, known as pro-inflammatory (PRO), and soluble diclofenac, a non-steroidal anti-inflammatory drug (ANTI), were intradiscally administered in a rat IVD injury model, 24 h after lesion. Two weeks after administration, a reduction of disc height accompanied by hernia formation was observed. In the PRO-inflammatory treated group, IL-1β, IL-6 and COX-2 IVD gene expression were upregulated, and loss of nucleus pulposus (NP) structure and composition was observed. Systemically, lower T-cell frequency was observed in the lymph nodes (LN) and spleen (SP) of the PRO group, together with an increase in CD4+ T cells subset in the blood (BL) and LN. In contrast, the ANTI-group had higher proteoglycans/collagen ratio and collagen type 2 content in the NP, while an increase in the frequency of myeloid cells, M1 macrophages and activated macrophages (MHCII+) was observed at the systemic level. Overall, this study illustrates the dynamics of local and systemic inflammatory and immune cell responses associated with intradiscal therapies, which will contribute to designing more successful immunomodulatory treatments for IVD degeneration.

## 1. Introduction

Inflammation is often related with intervertebral disc (IVD) degeneration, since a wide number of inflammatory mediators, including prostaglandins (PG), interleukins (IL-1, -6, -8) and tumor necrosis factor (TNF)-α have been described in IVD catabolic processes [1]. Conservative treatments often include anti-inflammatory administration [2,3], but the mechanisms that regulate inflammation associated with IVD degeneration are far from being fully understood.

Several in vivo models of IVD degeneration are described in the literature, with murine tail models of mechanical injury by needle puncture commonly used [4,5]. Depending on the needle size, the puncture may induce annulus fibrosus (AF) disruption while causing depressurization of the nucleus pulposus (NP) [6,7]. Needle calibers, varying between 16 to 22 G, often lead to significant pressure failure, decreased cell viability, increased expression of pro-inflammatory/degenerative factors and alterations in extracellular matrix (ECM) composition in IVDs from both large (e.g., bovine [8]) and small animals (e.g., rabbit [9] and rat [10]), resembling human IVD degeneration. Our group has previously established and validated an IVD herniation/degeneration model of rat caudal needle puncture, using a 21 G needle [10,11], in which IVD herniation and infiltration of macrophages were observed [10]. This model is relatively simple to manipulate and presents cost-effectiveness [12]. Moreover, it allows the study of local and systemic immune response, crucial in human IVD degeneration and difficult to analyze in vitro or ex vivo [13]. Only a few studies, mainly in bone, address the dialogue/interplay between local/systemic immune responses and how this can be linked with tissue remodeling [14,15]. Despite this, it is important to consider the limitations of small animal models of IVD degeneration when compared to humans, such as differences in spine biomechanics, IVD size and cellular composition, which may be critical for clinical translation [5].

Here, we hypothesize that pro- and anti-inflammatory intradiscal injections modulate local and systemic inflammatory responses associated with IVD herniation. To prove the hypothesis, intradiscal administration of PRO-inflammatory chitosan/poly-**γ**-glutamic acid nanocomplexes (Ch/**γ**-PGA NCs), previously developed by our group [16], was confronted with a nonsteroidal ANTI-inflammatory drug widely used in the clinics, diclofenac (Df). Ch/**γ**-PGA NCs were previously shown to significantly increase the production of IL-6 and TNF-α by macrophages [16] and dendritic cells [17] and to re-educate tumor macrophages towards a pro-inflammatory profile, decreasing the expression of CD163 and promoting the secretion of IL-12p40 and TNF-α [17]. On the other hand, Df treatment of IL-1β-stimulated NP organ cultures significantly downregulated IL-6 and IL-8 expression and PGE2 production [18]. In this study, we established the dynamics of systemic vs. local immune response to these stimuli in disc degeneration/herniation.

## 2. Results

The rationale of this study was to modulate the inflammatory/immune response associated with IVD degeneration and herniation. Animals with injured discs were treated with intradiscal injection of pro-inflammatory NCs solution (PRO) or Diclofenac (ANTI). The study design is illustrated in Figure 1.

### 2.1. Comparison of Intradiscal Pro-/Anti-Inflammatory Treatments in Disc Height Index and Local Inflammatory Response

The first effects of the intradiscal pro-/anti-inflammatory treatments were evaluated by disc height index (DHI) (Figure 2A). The results showed a decrease in percentage of DHI in all injured groups compared to noninjured animals, being significantly lower in PRO compared to VEHICLE (Figure 2B, *p* = 0.018).

Local IVD inflammatory response was evaluated by gene expression analysis of proinflammatory markers IL-1β, COX-2 and IL-6 (Figure 2C). These markers were either not expressed or expressed at low levels in INJURY, VEHICLE and ANTI groups. In the PRO group, it was observed an upregulation of IL-1β (*p* = 0.072), COX-2 (*p* = 0.045) and IL-6 (*p* = 0.040) in comparison to VEHICLE, and an upregulation of IL-1β (*p* = 0.039), COX-2 (*p* = 0.003) and IL-6 (*p* = 0.063) versus INJURY.

### 2.2. Comparison of Intradiscal Pro-/Anti-Inflammatory Treatments in NP ECM

To evaluate whether the modulation of inflammatory response impacts IVD tissue remodeling, histological analysis was performed on the NP and hernia for proteoglycans and collagen quantification. Representative images of the central NP sections for all groups are depicted in Figure 3Aa–e). Of note is that 1/6 animals from the INJURY group completely lost proteoglycans and NP integrity, and 4/6 animals from the PRO group did not present a normal NP proteoglycan structure. NP proteoglycans area was significantly reduced in all groups except the VEHICLE in comparison to NAÏVE animals. The highest proteoglycans/collagen ratio was observed in the ANTI group, while the PRO group present the lowest proteoglycans/collagen ratio (Figure 3A). A similar trend was observed for the percentage of COL2 in the NP (Figure 3B). In NAÏVE animals, no CD68+ macrophages were found. The percentage of CD68+ macrophages present in the NP of PRO-treated animals was higher compared to VEHICLE (Figure 3C, *p* = 0.042) and the ANTI group (*p* = 0.034), suggesting that the presence of these cells can be related with lower proteoglycan content in the NP (Figure 3A). Representative images of COL2 and CD68 immunostainings for all the conditions are provided in Figure 3B and Figure 3C, respectively.

### 2.3. Comparison of Intradiscal Pro-/Anti-Inflammatory Treatments in Disc Herniation

In this model, a pronounced hernia was formed upon injury, identified by extrusion of proteoglycan-rich tissue, occurring, in most cases, at the region between dorsal segmental muscles, as previously shown [10]. The herniated tissue was identified by Alcian blue/Picrosirius red staining (Figure 4Aa–d), and the volume was quantified by area delimitation of the hernia in sequential histological sections throughout the entire IVD, as described [10]. A significant reduction of hernia volume after intradiscal ANTI-inflammatory injection was observed compared to INJURY (*p* = 0.048), and in the proteoglycans/collagen ratio in PRO versus INJURY (*p* = 0.030). Since hernia resorption has been linked to macrophages’ activity [19], macrophage infiltration in the hernia was also assessed. Macrophages were found in hernias of all injured animals (Figure 4B), but not in healthy discs [10]. The percentage of CD68+ cells within the hernia was slightly decreased in the ANTI group compared to INJURY (*p* = 0.056), with ANTI the group having the smallest hernias, in accordance with previous reports [10]. This correlation suggests a better outcome in terms of hernia progression when an anti-inflammatory treatment is applied.

CCR7+ vessels were analyzed as a marker of inflammation within the hernia. CCR7 expression was found only on the border and immediate surroundings of the herniated tissue (Figure 4C). Number, diameter and thickness of vessel wall were quantified. No effect of the VEHICLE was observed in CCR7+ vessels, compared to the INJURY group. Although no differences were found to INJURY and VEHICLE, a higher thickness of CCR7+ vessels was observed in the PRO versus ANTI group (*p* = 0.005), suggesting an increase in the inflammatory status with the Ch/γ-PGA NCs administration. Representative images of CD68 and CCR7 immunostainings for all the conditions are provided in Figure 4B and Figure 4C, respectively.

### 2.4. Comparison of Intradiscal Pro-/Anti-Inflammatory Treatments in Systemic Immune Response

At a systemic level, immune cell populations were analyzed in blood (BL), draining lymph nodes (LN) and spleen (SP) of the different animal groups (Figure 5A). The ANTI group showed a significant increase compared to VEHICLE-injected animals in the percentage of myeloid cells (CD11b+TCR-), M1 (CD11b+CD40+) and activated macrophages (CD11b+MHCII+) in BL (*p* = 0.013, *p* = 0.080, *p* = 0.038, respectively), LN (*p* = 0.008, *p* = 0.006, *p* = 0.010, respectively) and SP (*p* = 0.010, *p* = 0.010, *p* = 0.004, respectively). On the other hand, a significant decrease in T cell population (TcR+CD161-) was observed in PRO-treated animals compared with VEHICLE in the SP (*p* = 0.038), accompanied by an increase in CD4+ T cells subset (CD4+CD8- in TcR+CD161- cells) in BL and LN (*p* = 0.028, *p* = 0.042, respectively). Surprisingly, T-cell activation (TcRDim in TcR+ cells) was significantly increased in VEHICLE versus INJURY in BL and LN (*p* = 0.004, *p* = 0.019, respectively). B cells population (CD45R+) was significantly increased in the SP of PRO versus VEHICLE (*p* = 0.004), while the percentage of cells expressing MHCII (activated antigen-presenting cells) was similar between the different groups for the analyzed tissues.

Additionally, the systemic profile of inflammation-related cytokines was analyzed using a cytokine array (Figure 5B). Interestingly, the results indicated that the NCs treatment (PRO group) induced less secretion of pro-inflammatory molecules, such as Fas ligand, IL-1β, IL-13, platelet-derived growth factor (PDGF)-AA, TNF-α and vascular endothelial growth factor (VEGF), in contrast to the ANTI-inflammatory treatment. To validate the array, IL-1β, IL-6 and PGE2 were quantified by ELISA in the animals’ plasma (Figure 5C). IL-1β and IL-6 concentrations were significantly lower in the PRO group versus INJURY (*p* = 0.029, *p* = 0.045, respectively).

## 3. Discussion

In the current study, local and systemic inflammatory responses of ANTI- versus PRO-inflammatory intradiscal injection in a rat model of IVD lesion were analyzed 2 weeks postlesion. This time point was selected based on our previous studies in which a larger IVD herniation was observed compared with 6 weeks post-lesion [10]. Anti-inflammatory treatment was based on Df, an FDA-approved anti-inflammatory drug for the treatment of low back pain. Df was previously intradiscally administered in a pro-inflammatory ex vivo model of bovine IVD, downregulating pro-inflammatory gene expression of IVD cells [18]. The PRO-inflammatory stimulus used Ch/**γ**-PGA NCs, since they were previously shown to activate macrophages [16] and stimulate tumor-associated macrophages and dendritic cells, thus being explored as adjuvant immunotherapy [17,20]. The treatment was based on a single intradiscal injection, to minimize tissue disruption, since IVD puncture (for example by discography) was, until recently, considered a possible cause of IVD degeneration [21].

In our study, Df (0.15 mg/kg) decreased IVD herniation compared with INJURY animals. At this time point, it seems that hydration of the disc by itself (upon vehicle injection) also contributes to reducing hernia volume, although not statistically significantly. This was unexpected, but an effect of vehicle intradiscal injection has been reported before; for example, phosphate-buffered saline (PBS) and TNF-α intradiscal injections induced similar IVD degeneration [22]. Nonetheless, despite Df’s short biologic half-life, it can be envisaged for local control of acute inflammation. Df may be incorporated into a delivery system, for a prolonged release kinetics, that may allow the use of lower doses than those prescribed for oral administration. Although rapidly released at physiological pH [16], Df has been previously incorporated in Ch/**γ**-PGA NCs with promising results in an ex vivo model [23]. Df has been intravenously administered to arthritic rats (10 mg/kg), decreasing systemic PGE2 plasma levels up to 360 min after administration [24]. In addition, Df has also been intraperitoneally administered in a rat lumbar disc herniation model (10 mg/kg), demonstrating a reduced analgesic effect with time [25].

Intradiscal injection of steroids has been shown to improve pain and disability in degenerative disc disease patients with end-plate changes, but with only minimal temporary improvement in patients without changes [26]. Moreover, epidural steroid injection was shown to decrease pain symptoms in patients with lumbar-herniated NP, but a similar decrease in hernia size was observed in the group without intradiscal treatment [27]. On the other hand, nonsteroidal anti-inflammatory drugs, such as Df, are considered the most effective in inhibiting COX and lipoxygenase pathways, decreasing inflammation and pain [28,29], and have been widely used in osteoarticular disorders [30].

An interplay between IVD inflammatory response and ECM remodeling has been previously suggested. The ANTI group shows a trend to increase proteoglycans and COL2 in the NP. This is in agreement with the literature, where Matta et al. [31] observed that intradiscal injection of recombinant transforming growth factor (TGF)-β1, an anti-inflammatory molecule, and connective tissue growth factor, increased the expression of aggrecan, COL2, Brachyury and octamer-binding transcription factor 4, in a rat-tail IVD model of needle puncture, 6 weeks after injury. In the present work, a decrease in DHI, as well as loss of proteoglycan content and of NP-AF border integrity were observed in all groups. Although **γ**-PGA and Ch/**γ**-PGA NCs were shown to promote cartilaginous ECM production in vitro and ex vivo [32,33], this was not confirmed in vivo, possibly due to the much higher injection/NP volume ratio in the rat than in the bovine [34]. We also cannot exclude the presence of the immune system in vivo, which was absent in previous studies [34]. Moreover, we cannot exclude the dynamics of inflammation and tissue remodeling at different timepoints. Cuellar and colleagues detected the highest production of IL-6, 3 h after inflammation induction, of IL-1β, at 6 to 24 h and of TNF-α, at 24 h in the epidural space of a rat model of noncompressive disc-herniation-induced inflammation [35]. In a rat tail torsion loading study, an overall downregulation of IL-1β, but not of IL-6 or TNF-α, was reported for the different conditions tested, compared to sham, 24 h after loading [36]. Moreover, MacLean et al. [37] demonstrated that gene expression levels of most catabolic and anabolic genes reach maximum levels 24 h following mechanical stimulation, although some peaked at 8 or 72 h following dynamic compression in caudal motion segments in vivo. Therefore, it would be interesting to analyze the inflammatory response at earlier time points after injury (up to 72 h). On the other hand, ECM protein production should be evaluated at later time points (4–10 weeks), as suggested by other works in vivo [31,38,39].

Interestingly, local inflammatory response seems to contrast with the systemic results, in which the analysis of the main immune cell populations and inflammatory cytokines profile showed significantly increased frequencies of myeloid cells, M1 and activated macrophages in BL, LN and SP of animals treated in the ANTI group, while the PRO group presented fewer T cells, particularly in SP, and a significant increase in B cells. Previous work, in a femoral bone injury model, showed a reduction in proportions of myeloid cells and activated myeloid cells, in BL, SP and LN, accompanied by increased T cells, but decreased activated T cells, 6 days after femoral bone injury (end of acute inflammation stage), in animals that progressed faster towards bone repair [14]. On the other hand, in a model of chronic inflammation, Collagen-Induced Arthritis (CIA), the proportion of activated myeloid cells was increased in SP 3 days after femoral bone injury [40]. These changes in cell proportions will be influenced by the specificities of the models and the time postinjury of the analysis, and in these studies, the local immune cell presence was not evaluated. In the same model of IVD degeneration, previous work showed that injury, with or without systemic MSC transplantation, resulted in lower proportion of myeloid cells in LN, with increased MHC-II positive proportions in blood and decreased in SP for the MSC-treated group (myeloid and B cells together), but the local analysis showed no increase in monocytes at the hernia [11]. These changes in systemic cell proportions may reflect the cell recruitment to the place of injury, which is supported by the histological results presented here, showing increased levels of myeloid CD68+ cells in the NP and hernia of the PRO group. Nonetheless, a detailed analysis of immune cell recruitment to herniated IVD is still lacking. Since an antigen-specific immune response has been considered in case of hernia regression [41], it would be relevant in the future to follow that immune cell recruitment to the injury site by analyzing the cellular content in the IVD, hernia, BL, LN and SP at different time points, namely in the acute inflammatory response. Our results also evidence the importance of looking to both local and systemic inflammatory and immune cell responses in vivo to better understand the dynamic nature of the immune response to IVD herniation. Although it is known that NP tissue contact with systemic circulation leads to increased immune cell activation and cytokine production [42], only a few studies have analyzed the immune peripheral response, along with the local inflammatory status of the disc. Akyol and colleagues [43] found that changes in the levels of IL-17, TGF-β, IL-6 and IFN-γ in degenerative disc samples have been similarly reflected in the peripheral blood. In parallel, some studies involving low back pain patients have tried to address how systemic inflammation correlates with disease prognosis [44,45]. Higher IL-6 systemic levels in patients with lumbar disc herniation were correlated with poorer recovery, while a decrease in systemic IL-6 was correlated with improved pain in these patients [44]. Treatment with epidural injection of steroids reduced systemic IL-17 and VEGF in patients with disc herniation [44]. The systemic inflammatory response to PEEK particulate debris administered intradiscally in a rabbit in vivo model showed a significant increase in systemic TNF-α and IL-1β after 3 months that was maintained until 6 months [46]. The maintenance of pro-inflammatory systemic markers in the plasma was associated with the non-biodegradability of PEEK. Looking into our work, the low levels of systemic pro-inflammatory cytokines suggest that Ch/γ-PGA nanoparticles (PRO-inflammatory treatment) were degraded within 2 weeks. γ-PGA is degraded by g-Glutamyl-transpeptidases (GGT) into D-/L-glutamate units [47]. Ch is known as non-biodegradable, being able to drive the inflammatory response from pro- to anti-inflammatory with time, as we have previously described when culturing monocytes in Ch ultra-thin films, with an increase in TNF-α and IL-1β secretion after 3 days, that switch to an increase in IL-10 and TGF-β upon 10 days [48]. Regarding the systemic inflammatory response to the ANTI-inflammatory treatment, although Df is an NSAID extensively used as a COX inhibitor, its true mechanism of action remains to be highlighted [49]. In a clinical follow-up study of the systemic inflammatory response upon aneurysmal subarachnoid hemorrhage (SAH), the authors verified an inverse correlation between the amount of NSAID applied and the systemic pro-inflammatory parameters (IL-6 and C-Reactive Protein) in the acute phase (≤day 14) [50].

## 4. Materials and Methods

### 4.1. Animal Experimentation

Male Wistar Han (Crl:WI/Han) rats (36 animals, *n* = 6/group, age = 2 months, weight = 313 ± 27 g) were used for the IVD caudal injury model as previously established [10]. Briefly, the animals were anesthetized by isoflurane inhalation and placed in prone position. A 21G percutaneous puncture was performed in three consecutive coccygeal IVDs (Co5/6, Co6/7 and Co7/8), using radiography for disc identification.

After 24 h, lesioned animals were intradiscally administered with 10 µL of: (1) soluble Df [18] (ANTI); (2) Ch/**γ**-PGA NCs [17] (PRO) and (3) NCs buffer: 0.05M Tris-HCl and 0.15 M NaCl [51] (VEHICLE). Injured-only animals were kept as controls (INJURY). A 33 G needle coupled to a microsyringe (Hamilton, Reno, Nevada, EUA) and an adaptor to assure 5 mm depth administration to the center of the IVD were used. Each animal received one treatment in three consecutive discs. Polymers’ and NC’s preparation are described elsewhere [51,52], NCs characterization can be found in Appendix A (Figure A1A) and NCs concentration used was 35 mg/mL (approximately 1.75 mg/kg). Df sodium salt (Sigma-Aldrich, St. Louis, MO, EUA) concentration used was 3 mg/mL (approximately 0.15 mg/kg), estimated based on the doses used in previous work in bovine IVD [16,18], intravenous (2 mg/kg) [53] or intraperitoneal (10 mg/kg) injections [25].

The timepoint of administration (24 h) was selected based on IL-1β and PGE2 systemic profiles, which peaked in the acute phase of the inflammatory response (Appendix B, Figure A2). Two weeks later, the animals were sacrificed for tissue collection (Figure 1). Locally, disc height index (DHI), gene expression of pro-inflammatory markers (IL-1β, COX-2, IL-6) by IVD cells, proteoglycan and collagen content, hernia size, presence of CD68+ and CCR7+ cells by histology and immunohistochemistry were determined. Systemically a profiling was performed of immune cell populations in blood (BL), draining lymph nodes (LN) and spleen (SP) by flow cytometry. Inflammatory cytokines in plasma were identified by protein array and ELISA.

All experiments were carried out at i3S—Instituto de Investigação e Inovação em Saúde animal facility, in accordance with European Legislation on Animal Experimentation through the Directive 2010/63/UE and approved by the Institute’s Animal Ethics Committee and Direcção Geral de Alimentação e Veterinária through the license no. 3773/2015-02-09.

### 4.2. Determination of the Disc Height Index

Digital radiographs were acquired by the Owandy-RX radiology system equipped with an Opteo digital sensor and processed with QuickVision software (Owandy Radiology, Croissy-Beaubourg, France). The disc height index (DHI) was calculated as shown in Figure 2, using ImageJ software for radiograph measurements, as previously described [10].

### 4.3. RNA Isolation and Quantitative Real-Time qPCR

Total RNA was isolated from the NP using TRIzol reagent (Invitrogen, Carlsbad, CA, EUA) and quantified by Nanodrop spectrophotometry (Thermo Fisher Scientific, Waltham, MA, EUA). Two NPs per animal were collected and pooled. Isolated RNA was treated with DNase (Thermo Fisher Scientific) and transcribed using the high-capacity cDNA reverse transcription kit (Applied Biosystems, Foster City, CA, EUA). Gene expression levels were determined by qPCR using the iQ5 Real-Time PCR Detection System (Bio-Rad Laboratories, Hercules, CA, USA), TaqMan Gene Expression Master Mix and Taqman Assays (Applied Biosystems) for interleukin (IL)-1β (Rn00580432_m1), IL-6 (Rn00561420_m1), cyclooxygenase (COX)-2 (Rn01483828_m1) and glyceraldehyde 3-phosphate dehydrogenase (GAPDH, Rn99999916_s1) as reference gene. Quantification cycle (Cq) 35 cutoff was used. Relative expression levels were calculated using the Cq method (∆Ct = Ct(gene of interest) – Ct(GAPDH)), according to published guidelines [54].

### 4.4. IVD Collection and Histological Analysis

Two weeks postinjury, one IVD per animal was collected with adjacent vertebrae and fixed in 10% neutral buffered formalin for 1 week at room temperature. Tissue was decalcified in EDTA–glycerol solution and processed for paraffin embedding. Sequential transversal 5 µm sections of the IVD were collected. Sections were stained with Alcian blue/Picrosirius red staining throughout the IVD length to identify proteoglycans and collagen tissue distribution. Sections were imaged with light microscopy (CX31, Olympus, Tokyo, Japan). Hernia and NP areas were determined by delimitating regions of interest in each optical section, after image calibration, considering Alcian blue staining for proteoglycans and Picrosirius red staining for collagen. Hernia volume was calculated as the sum of areas of each individual section throughout the IVD as described [10]. Within hernia and NP regions of interest, the percentage area of proteoglycans and collagen was determined by a custom ImageJ macro based on a color deconvolution technique used to separate the color channels from Alcian blue and Picrosirius red and that allows quantification of the ECM content observed in the histological sections [55].

### 4.5. Detection of Collagen Type II in the IVD

Collagen type II (COL2) distribution in the IVD was analyzed by immunofluorescence staining. Antigen retrieval was performed in paraffin sections through incubation with 20 μg/mL proteinase K (Sigma-Aldrich) solution for 15 min at 37 °C. Sections were incubated overnight, at 4 °C, with anti-collagen II-II6B3 (1:20, Developmental Studies Hybridoma Bank at the University of Iowa, Department of Biology, Iowa City, IA, USA) antibody, followed by incubation with Alexa Fluor 594-labeled goat anti-mouse (1:1000, Invitrogen) antibody. Sections were mounted in Fluorshield with DAPI (Sigma-Aldrich). Representative images of the slides were taken using fluorescence microscopy (Axiovert 200 M, Zeiss, Oberkochen, Germany). The percentage of COL2 in NP was determined as described by Cunha et al. [11].

### 4.6. Detection of CD68+ and CCR7+ Cells

Immunohistochemistry against CD68+ and CCR7+ cells in the hernia was performed by the NovolinkTM Polymer Detection Kit (Leica Biosystems, Wetzlar, Germany), following the manufacturer’s instructions. Antigen retrieval was performed through incubation in near-boiling point 10 mmol/L sodium citrate buffer, pH 6.0, for 1 min, followed by incubation with 20 μg/mL proteinase K solution for 15 min at 37 °C. Sections were incubated with anti-CD68-ED1 (1:100, Bio-Rad Laboratories) or anti-CCR7 (1:5000, Abcam, Cambridge, UK) antibodies overnight at 4 °C. The stained sections were imaged with light microscopy. CCR7+ positivity was quantified using ImageJ tools directly on the acquired images, and CD68+ images were subjected to ImageJ color deconvolution technique to separate DAB and hematoxylin channels [55]. From these, the area of CD68 positivity was normalized to the area of hematoxylin to obtain the percentage of CD68+ cells.

### 4.7. Flow Cytometry Analysis of Systemic Immune Cell Populations

After intracardiac exsanguination under isoflurane anesthesia, blood (BL), draining lymph nodes (LN) and spleen (SP) were collected and immediately processed for flow cytometry analysis. Whole blood was collected in 1:10 anticoagulant citrate–phosphate–dextrose solution with adenine (Sigma-Aldrich). After dissection, the iliac and inguinal LN and the SP were collected in Roswell Park Memorial Institute (RPMI) 1640 medium supplemented with 10% fetal bovine serum to maintain cell viability until processing.

Peripheral BL mononuclear cells were isolated by centrifugation over lymphoprep (Axis-Shield, Dundee, UK) at 800 g, for 30 min without brake, at room temperature. Plasma was collected, spun to remove cell debris and kept at −80 °C for cytokine analysis. Cells were collected from the interface between plasma and lymphoprep. LN cells were isolated by mechanical dissociation of the LN. SP cells were collected by a similar process after enzymatic digestion with 100 U/mL Collagenase type I (Sigma-Aldrich).

Cell surface staining for flow cytometry was performed in 96-well plates in buffer (PBS, 0.5% BSA, 0.01% sodium azide) for 30 min on ice, after Fc receptors blocking. The following anti-rat antibodies (BD Pharmingen, San Diego, CA, USA), were used: CD45R-PE (clone HIS24), TCR-PerCP (clone R73), major histocompatibility complex class II (MHCII)-PerCP (clone OX-6), CD4-APC (clone OX35), and CD161a-FITC (clone 10/78), CD11b/c-PE-Cy7 (clone OX-42) and CD8-V450 (clone OX-8), CD40-FITC (clone HM40-3). Samples were acquired on a flow cytometer (FACSCanto II, BD), and data were analyzed with FlowJo software version 8.7 (FlowJo LLC, Ashland, OR, USA).

### 4.8. Plasma Cytokine Quantification

The membrane-based Rat Cytokine Antibody Array C2 (RayBiotech, Norcross, GA, USA) was used for semi-quantitative detection of cytokines in the plasma according to manufacturer’s instructions. The plasmas pool (*n* = 6) for the groups INJURY, PRO and ANTI was analyzed. Signal density for each sample spot was determined using Chemidoc XRS+ (Bio-Rad Laboratories) and ImageJ software. Relative cytokine levels were normalized to positive internal control and to the INJURY group. Interferon (IFN)-**γ**, macrophage inflammatory protein (MIP)-3α and IL-1β were not detected in the INJURY group, so for these, an arbitrary value of 1.5 instead of 0 was considered for calculation of fold-change. PGE2 (Arbor Assays, Ann Arbor, MI, USA), IL-1β and IL-6 (PeproTech, London, UK) were quantified by ELISA in the animals’ plasma as per the manufacturers’ instructions.

### 4.9. Statistical Analysis

Gene expression results are shown in dot plots. The remaining results are presented in box-and-whisker plots; the box indicates median ± interquartile range, and the whiskers represent a 95% confidence interval. Normality was assessed by D’Agostino-Pearson omnibus normality test. Statistical analysis was performed with non-parametric unpaired Kruskal-Wallis test, followed by Dunn’s multiple comparison test, using GraphPad Prism 7. Statistical significance was set at *p* < 0.05.

## 5. Conclusions

Overall, this study gives new pieces of evidence on the inflammatory response in degenerated IVD. We have shown that a pro-inflammatory intradiscal administration resulted in a more severe IVD lesion, translated at the systemic level by lower T-cell numbers and pro-inflammatory cytokines, whereas intradiscal anti-inflammatory diminishes the degradation process of the lesioned IVD, with increased levels of myeloid cells systemically. Therefore, this study contributes with insights to the dynamics of systemic inflammatory response associated with IVD degeneration, and to design more effective therapies to promote IVD regeneration.

## Figures and Tables

**Figure 1 ijms-21-01730-f001:**
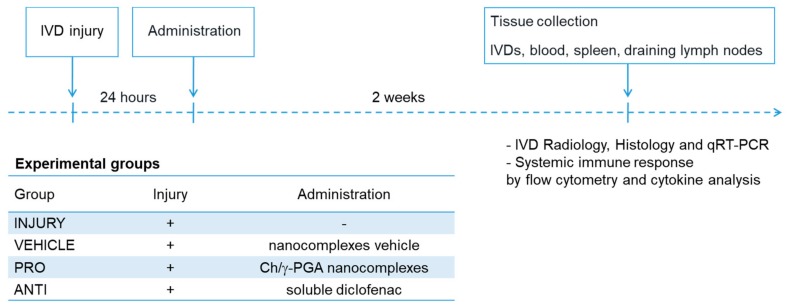
Experimental timeline.

**Figure 2 ijms-21-01730-f002:**
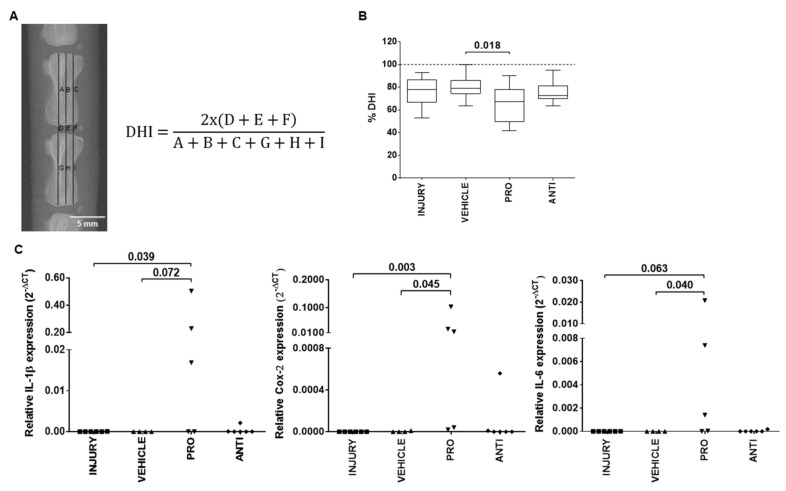
Local effect of the different injectable treatments (VEHICLE; PRO-; and ANTI-inflammatory) in the radiological features and local inflammatory response of degenerated/herniated intervertebral disc (IVD), 2 weeks postinjury. (**A**) Representative digital radiograph and disc height index (DHI) calculation formula obtained as the mean of three measurements from midline to the boundary of the central 50% of disc width, divided by the mean of the two adjacent vertebral body heights. (**B**) Percentage of DHI (% DHI) calculated for each disc by the difference in DHI between post- and preinjury. The % DHI preinjury corresponds to 100% (dashed line). Results are presented in box-and-whiskers plots with associated *p*-value (*n* = 6 mice/group). (**C**) Relative gene expression of pro-inflammatory markers IL-1β, COX-2 and IL-6 in lesioned IVDs, normalized to GAPDH. Results are presented in dotplots with associated *p*-value (*n* = 4–6 animals/group).

**Figure 3 ijms-21-01730-f003:**
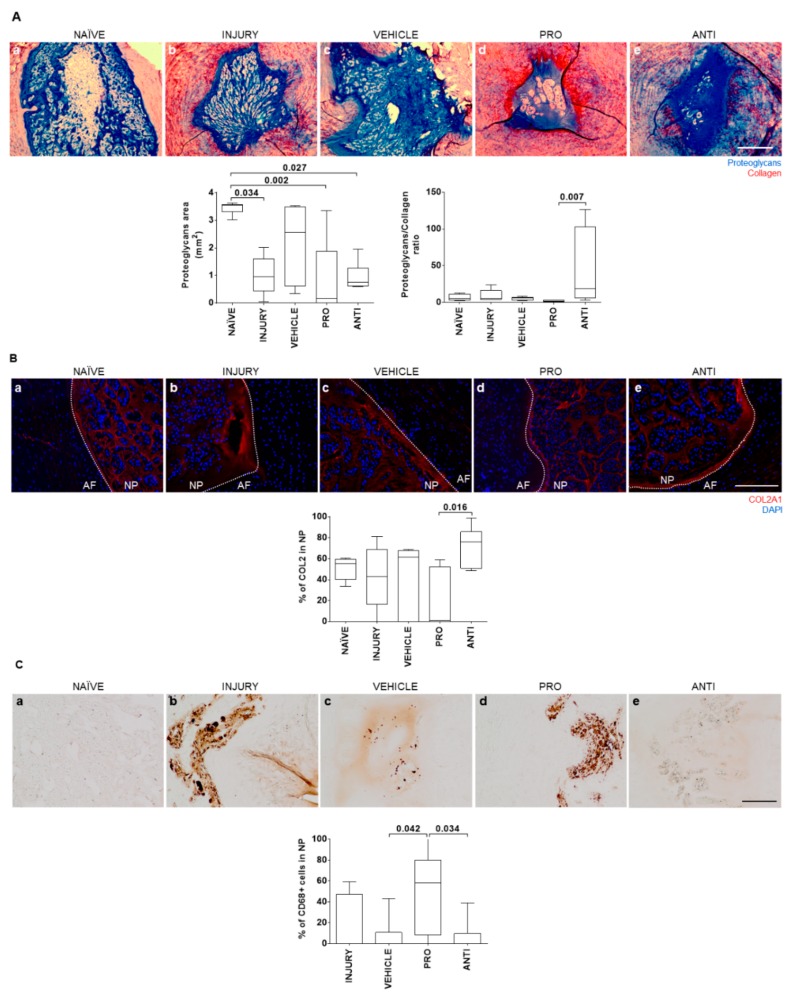
Histopathological analysis of nucleus pulposus (NP) tissue in naive and lesioned IVDs 2 weeks post-injury and intradiscal administrations. (**A**) Representative images of NP extracellular matrix (ECM) by Alcian blue/Picrosirius red staining (proteoglycans in blue and collagen in red; scale bar, 500 µm); quantification of the proteoglycans area (mm^2^) and proteoglycans/collagen ratio in the NP. (**B**) Representative images of collagen type II (COL2) staining in the NP for all groups (COL2 is stained in red and DAPI stains cell nuclei in blue; scale bar, 200 µm); percentage of COL2 area in the NP. (**C**) Representative images of macrophages in the NP by CD68 immunostaining for all groups (positive cells are stained in brown; scale bar, 200 µm); quantification of the percentage of CD68+ cells in the NP. Results are presented in box-and-whisker plots with associated *p*-value (*n* = 6 animals/group).

**Figure 4 ijms-21-01730-f004:**
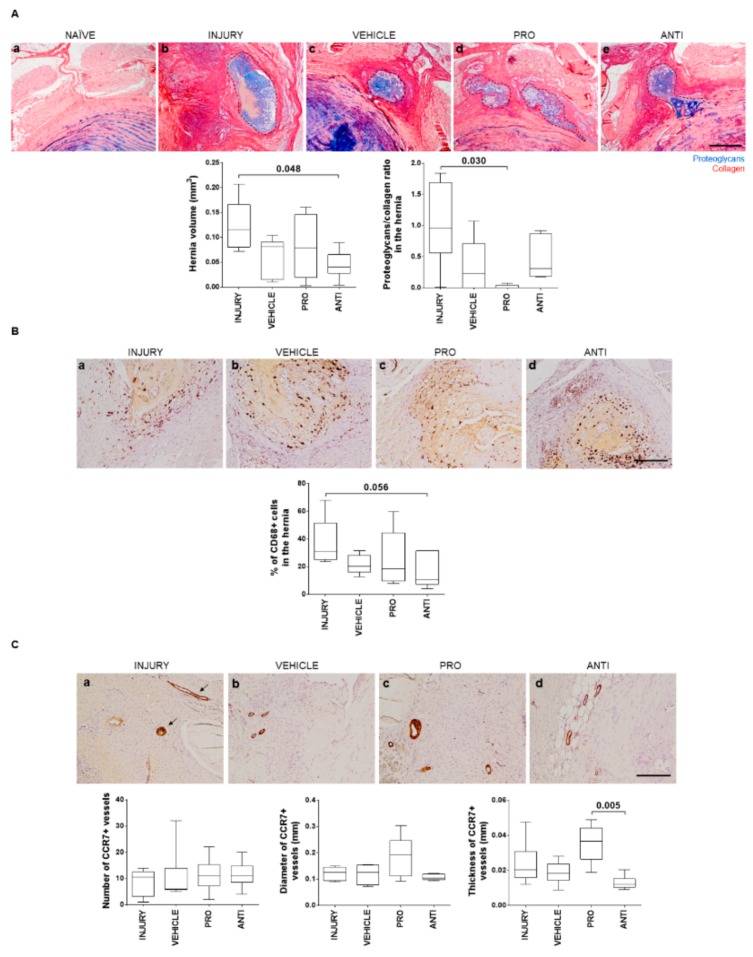
Histopathological analysis of herniation regions in lesioned IVDs 2 weeks post-injury and intradiscal administrations. (**A**) Representative images of the hernia formed in the different animal groups by Alcian blue/Picrosirius red staining (hernia delimited by dashed line; scale bar, 500 µm); hernia volume (mm^3^) was calculated from the staining across the depth of all sections of an IVD with visible herniation. (**B**) Macrophages identification within the hernia by CD68 immunostaining (positive cells are stained in brown). Representative images for all groups and respective CD68+ quantification within the hernia (scale bar, 200 µm). (**C**) Blood vessels immunoreactivity to CCR7. Representative image of CCR7+ immunostaining for all groups. Blood vessels are stained in brown (arrows; scale bar, 200 µm). Quantification of number, diameter (mm) and thickness (mm) of vessels. Results are presented in box-and-whiskers plots with associated *p*-value (*n* = 6 animals/group).

**Figure 5 ijms-21-01730-f005:**
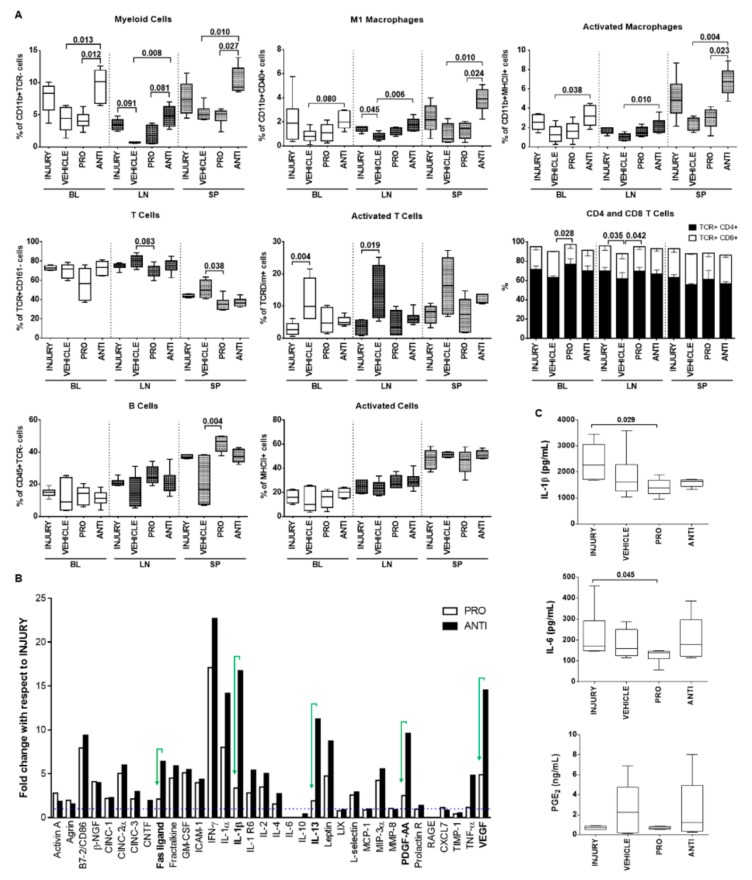
Systemic immune response of animals with degenerated/herniated IVDs injected with pro- and anti-inflammatory treatments 2 weeks postinjury. (**A**) Phenotypic analysis of immune cell populations in peripheral organs: blood (BL), draining lymph nodes (LN) and spleen (SP), using flow cytometry. (**B**) Semiquantitative antibody detection of an array of inflammatory mediators in the rat plasma for INJURY, PRO and ANTI groups, expressed as fold-change with respect to the INJURY group (dashed line, y = 1). IFN-*γ*, MIP-3α and IL-1β were not detected in the INJURY group; for these, an arbitrary value of 1.5 was considered for calculation of fold-change for the remaining groups. (**C**) Quantification of pro-inflammatory cytokines IL-1β (pg/mL), IL-6 (pg/mL) and PGE_2_ (ng/mL) in animals’ plasma by ELISA. Results are presented in box-and-whisker plots with associated *p*-value (*n* = 4–8 animals/group), except the array results presented in a bar chart.

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
