# Peer review of "Modulation of the In Vivo Inflammatory Response by Pro- Versus Anti-Inflammatory Intervertebral Disc Treatments"

_ijms, 2020, doi:10.3390/ijms21051730_

Round 1
Reviewer 1 Report
This manuscript investigates the modulation of an inflammatory response in a rat IVD injury model after application of pro- vs anti-inflammatory substances. The topic is interesting and relevant for clinical practice as the long term perspective is the development of better therapies for the treatment of IVD degeneration.
The manuscript is well-written and structured. The methods used are well-described and appropriate. Experiments were performed systematically and the data presented are of high quality.
Even though the main purpose of this study was to compare pro- with anti-inflammatory treatments in an injury model it would make sense to include at least in some experiments an uninjured control. For all histological changes, it could be interesting to see how injury itself affects the structure, PGs etc and , more important, to what extent the injury induced changes can be reverted (back to uninjured controls). Also, it would be nice to show at least one representative image for each group in Figure 3B, C and Figure 4B, C. If it does not fit into the main ms than a supplemental figure could be provided.
Minor comments:
I understand that, in Figure 2B, %DHI indicates the differences in DHI pre- compared to post-operatively…if so, it should be added in the legend that pre=100%.
Line 56: for clinical translation instead of results translation?
Line 83: Figure 2C!
Author Response
This manuscript investigates the modulation of an inflammatory response in a rat IVD injury model after application of pro- vs anti-inflammatory substances. The topic is interesting and relevant for clinical practice as the long-term perspective is the development of better therapies for the treatment of IVD degeneration.
The manuscript is well-written and structured. The methods used are well-described and appropriate. Experiments were performed systematically and the data presented are of high quality.
Even though the main purpose of this study was to compare pro- with anti-inflammatory treatments in an injury model it would make sense to include at least in some experiments an uninjured control. For all histological changes, it could be interesting to see how injury itself affects the structure, PGs etc. and, more important, to what extent the injury induced changes can be reverted (back to uninjured controls). Also, it would be nice to show at least one representative image for each group in Figure 3B, C and Figure 4B, C. If it does not fit into the main ms than a supplemental figure could be provided.
The authors acknowledge the reviewer´s comments and have added data from uninjured controls to Figures 3 and 4. Representative images for COL2 and CD68 immunostaining in the NP (Figure 3) and CD68 and CCR7 immunostainings in the hernia site (Figure 4) were added to the manuscript.
Minor comments:
I understand that, in Figure 2B, %DHI indicates the differences in DHI pre- compared to post-operatively…if so, it should be added in the legend that pre=100%.
The authors have added the following sentence “The % DHI pre-injury corresponds to 100% (dashed line).” to the legend of Figure 2B.
Line 56: for clinical translation instead of results translation?
The authors have altered the sentence.
Line 83: Figure 2C!
The authors acknowledge the reviewer´s comments and have corrected the Figure 2 citation.
Reviewer 2 Report
This research manuscript demonstrates the dynamics of local and systemic inflammatory responses associated with intervertebral disc (IVD) herniation and intradiscal pro-/anti-inflammatory treatments of IVD herniation. As expected, the pro-inflammatory treatment was found to increase expression of pro-inflammatory markers, such as IL-1β, Cox-2 and IL-6, in lesioned IVD tissues. On the other hand, the anti-inflammatory treatment was found to promote regeneration of nucleus pulposus (NP) tissues, as indicated by higher proteoglycan/collagen ratio, higher percentage of collagen type 2 compared to that in pro-inflammatory treated NP tissues. Additionally, the anti-inflammatory treatment reduced percentage of macrophages in NP tissues and reduced inflammation within the herniation regions of lesion IVD tissues. At systemic level, the anti-inflammatory treatment increased percentage of myeloid cells in blood and increased percentage of myeloid cells and M1 macrophages in spleen. Additionally, the anti-inflammatory treatment enhanced secretion of inflammatory mediators, such as Fas ligand, IL-1β, IL-13, PDGF-AA and VEGF. On the other hand, the pro-inflammatory treatment enhanced secretion of pro-inflammatory cytokines, such as IL-1β and IL-6. Overall, the manuscript is interesting. However, there are some comments that the authors need to address.
1. The authors should explain why they applied only 1 dose of pro-/anti- inflammatory treatment on IVD herniation, and why they opted 2 weeks post-treatment as time point for the treatment response assessment.
2. Lines 79-81 and lines 82-83: The sentences in these lines were cited with the wrong figures. Please revise.
3. Figure 5A: The anti-inflammatory treatment was found to increase percentage of pro-inflammatory M1 macrophages in spleen. Please discuss.
4. Figure 5A: The vehicle treatment was found to increase percentage of activated T cells in blood and lymph node. Please discuss.
5. Figure 5B: The pro-inflammatory treatment induced less secretion of inflammatory mediators, such as Fas ligand, IL-1β, IL-13, PDGF-AA and VEGF, compared to that of the anti-inflammatory treatment. Please discuss.
Author Response
- The authors should explain why they applied only 1 dose of pro-/anti- inflammatory treatment on IVD herniation, and why they opted 2 weeks post-treatment as time point for the treatment response assessment.
The authors acknowledge this relevant comment. This study aims to compare the dynamics of local vs systemic inflammatory response upon intradiscal treatment. With this in mind we envisaged an intradiscal therapy relying on a single, and not multiple, injection(s) since IVD puncture (for example by discography) was, until recently, looked as a possible cause of IVD degeneration (Stout, Discography, Med Rehabil Clin N Am, 2010). Regarding the option for analyzing the animal model 2 weeks post-treatment, this was based in our previous study that showed a higher IVD herniation at 2 weeks post-injury, that spontaneously partially resorbs after 6 weeks (Cunha, J Orthop Res, 2017). Later on, we also used this time point to address the therapeutic effect of MSCs systemic injection (Cunha, Stem Cells Transl Med, 2017). We have highlighted both comments in the discussion of the manuscript.
- Lines 79-81 and lines 82-83: The sentences in these lines were cited with the wrong figures. Please revise.
The authors acknowledge the reviewer´s comments and have corrected the Figures citations.
- Figure 5A: The anti-inflammatory treatment was found to increase percentage of pro-inflammatory M1 macrophages in spleen. Please discuss.
We thank the reviewer for this comment, to address it in the revised version of our manuscript we discuss these results in view particularly of other results from our group using this model and a model of femoral injury (Pg 10, L 244). These results support the hypothesis that in pro-inflammatory treatment macrophages are recruited to the injury site, thus decreasing in systemic organs. However, a detailed study of macrophage recruitment to herniated IVD, is still lacking.
- Figure 5A: The vehicle treatment was found to increase percentage of activated T cells in blood and lymph node. Please discuss.
We thank the reviewer for this comment, to address it in the revised version of our manuscript we discuss these results in view particularly of other results from our group using this model and a model of femoral injury (Pg 10, L 244). The results in relation to the activated T cell proportion might be in the same line as the ones with macrophages, but as these populations require labelling for multiple membrane receptors their presence at local level was not evaluated in the scope of the current work.
- Figure 5B: The pro-inflammatory treatment induced less secretion of inflammatory mediators, such as Fas ligand, IL-1β, IL-13, PDGF-AA and VEGF, compared to that of the anti-inflammatory treatment. Please discuss
We acknowledge the reviewer’s comment. The systemic response to injury is characterized by an increase in the pro-inflammatory cytokines upon lesion, which is also associated with a lower ability for lesion repair. This is a highly dynamic process, starting immediately after trauma and lasting for some weeks/months until resolving. Also, not many studies addressed the systemic immune response to disc lesion and intradiscal treatments, but we have addressed this comment in the Discussion. It would be interesting in the future to understands the dynamics of the inflammatory response upon intradiscal treatment. Nevertheless, we discussed this point in the manuscript.
Round 2
Reviewer 2 Report
The quality of the manuscript has been significantly improved. I have no further comment.